# The Molecular Docking of MAX Fungal Effectors with Plant HMA Domain-Binding Proteins

**DOI:** 10.3390/ijms242015239

**Published:** 2023-10-16

**Authors:** Lina Rozano, James K. Hane, Ricardo L. Mancera

**Affiliations:** 1Curtin Medical School, Curtin Health Innovation Research Institute, GPO Box U1987, Perth, WA 6845, Australia; 2Curtin Institute for Data Science, Curtin University, GPO Box U1987, Perth, WA 6845, Australia; 3Centre for Crop and Disease Management, School of Molecular and Life Sciences, Curtin University, GPO Box U1987, Perth, WA 6845, Australia

**Keywords:** fungal effector proteins, plant HMA domain protein, protein–protein interactions, molecular docking

## Abstract

Fungal effector proteins are important in mediating disease infections in agriculturally important crops. These secreted small proteins are known to interact with their respective host receptor binding partners in the host, either inside the cells or in the apoplastic space, depending on the localisation of the effector proteins. Consequently, it is important to understand the interactions between fungal effector proteins and their target host receptor binding partners, particularly since this can be used for the selection of potential plant resistance or susceptibility-related proteins that can be applied to the breeding of new cultivars with disease resistance. In this study, molecular docking simulations were used to characterise protein–protein interactions between effector and plant receptors. Benchmarking was undertaken using available experimental structures of effector–host receptor complexes to optimise simulation parameters, which were then used to predict the structures and mediating interactions of effector proteins with host receptor binding partners that have not yet been characterised experimentally. Rigid docking was applied for both the so-called bound and unbound docking of MAX effectors with plant HMA domain protein partners. All bound complexes used for benchmarking were correctly predicted, with 84% being ranked as the top docking pose using the ZDOCK scoring function. In the case of unbound complexes, a minimum of 95% of known residues were predicted to be part of the interacting interface on the host receptor binding partner, and at least 87% of known residues were predicted to be part of the interacting interface on the effector protein. Hydrophobic interactions were found to dominate the formation of effector–plant protein complexes. An optimised set of docking parameters based on the use of ZDOCK and ZRANK scoring functions were established to enable the prediction of near-native docking poses involving different binding interfaces on plant HMA domain proteins. Whilst this study was limited by the availability of the experimentally determined complexed structures of effectors and host receptor binding partners, we demonstrated the potential of molecular docking simulations to predict the likely interactions between effectors and their respective host receptor binding partners. This computational approach may accelerate the process of the discovery of putative interacting plant partners of effector proteins and contribute to effector-assisted marker discovery, thereby supporting the breeding of disease-resistant crops.

## 1. Introduction

Fungal effector proteins mediate the infection and disease symptoms of agriculturally important crops. During infection, phytopathogenic fungi secrete effector proteins into the host, which localise into the plant cell cytoplasm or the apoplastic space. Effector proteins are small, secreted proteins that are usually high in cysteine residues and less than 200 amino acids in length. The recognition of these effectors by the host may trigger downstream host defence mechanisms, and the type of effector–receptor interaction that takes place will depend on the type of effector and plant receptor that it interacts with.

Fungal pathogens rely on different effector sub-types—avirulence proteins (Avr) or necrotrophic effectors (NEs)—depending on their infective lifestyle, which may be biotrophic, hemi-biotrophic or necrotrophic [1]. Both Avrs and NEs are recognised by cognate host receptor proteins with nucleotide-binding site and leucine-rich repeat (NBS-LRR) domains. Avr interacts with Avr-resistance (R) receptor proteins, in a ‘gene-for-gene’ manner, where the host is resistant to an Avr if its cognate R receptor is present. Conversely, NEs interact with susceptibility (S) receptors in an ‘inverse gene-for-gene’ manner, where the host can become resistant to an NE if its cognate S receptor is lost [2]. Knowledge of both Avr-R and NE-S interactions is thus crucial in disease resistance breeding programs, which allow breeders to identify specific R/S-genes to be isolated and introduced into or removed from commercially bred cultivars that are resistant to disease. The same applies to identifying susceptibility factors and searching for alleles of these factors that would result in a weaker interaction with the effector, making the corresponding plant less susceptible to infection.

There are two modes of interaction of Avr-R/NE-S. The first is the indirect recognition of effectors by R/S proteins through intermediate binding proteins, such as the interaction of effector AvrPib with the wheat R protein Pib [3], Avr2/SIX3 (*Fusarium oxysporum* f. sp. *lycopersici*) with I-2 (tomato) [4,5], ToxB (*Pyrenophora tritici repentis*) with Tsc2 (wheat) [6,7,8], and AvrP (*Melampsora lini*) with P protein (flax) [9,10]. The second is the direct interaction of Avr and NE with plant R and S proteins, respectively. Unlike in the first mode, direct protein–protein interactions can be measured experimentally using yeast-2-hybrid (Y2H) assays [11], co-immunoprecipitation (co-IP) [11], biomolecular fluorescence complementation (BiFC) [12], or the determination of the 3D structure of bound complexes using nuclear magnetic resonance (NMR) [13] or X-ray diffraction (XRD) [14]. Both modes of interaction are equally of interest to plant breeders since at both a genetic and phenotypic level it does not impact the breeding process if the interaction is direct or indirect, and basic knowledge of effector interactions can accelerate breeding for resistance or susceptible gene traits of the host against fungal diseases [15]. From a practical point of view, however, in considering the application of structural bioinformatics to predicting novel effector–receptor interactions to support breeding, the second direct mode is the most feasible. Previously, the early discovery of Avr-R/NE-S interactions has relied on genetic mapping or quantitative trait locus (QTL) studies, which can be laborious and time-consuming [16]. Recently, effector discovery has been improved using genome-wide association single nucleotide polymorphism (SNP) studies [17].

Computational approaches offer an alternative for the prediction of Avr-R/NE-S interactions that can be validated experimentally. Advances in sequence-based bioinformatics methods for the high-throughput identification of effectors have predicted a large number of effector candidates from the secretomes of several phytopathogenic fungi [18,19,20]. However, many predicted candidates are false positives and, hence, new methods for filtering and prioritising effector candidates on the basis of predicted interactions with a specific host receptor binding partner are timely and broadly applicable. Molecular docking simulations can be utilised to investigate Avr-R/NE-S interactions where there is a direct interaction. There are two modes of docking. In rigid docking, one protein is kept stationary whilst the other (usually smaller) protein can move and rotate with respect to the first during sampling, which is implemented in docking programs such as ZDOCK [21] and ClusPro [22]. In flexible docking, a level of molecular flexibility is allowed, whereby both proteins are free to move and rotate during sampling, which is implemented in docking programs such as HADDOCK [23], FRODOCK [24], and SwarmDock [25].

For a study of the interaction between effectors and plant proteins (Avr-R/NE-S), the 3D structures of both proteins are required, which can be obtained from the Protein Data Bank (PDB), a database that contains all publicly available experimentally validated 3D structures of proteins resolved using XRD, NMR, or electron microscopy (EM) studies [26]. The main challenge in studying the interaction between effector and plant protein is the extremely limited number of available experimentally validated 3D structures of effector proteins. Up to the date of the completion of this work, there are 28 structures of effectors (including variants) in the PDB, and only seven of them, all from the MAX effector structural family, are in complex with their host receptor binding partner. Hence, this study focusses exclusively on the binding interactions of the MAX effector family. The MAX effector family includes multiple effectors from *Magnaporthe oryzae* and the ToxB of *Pyrenophora tritici-repentis. M. oryzae* has five MAX effectors whose structures are known (Avr1CO39, AvrPia, AvrPib, AvrPik, and AvrPiz-t) and which lack sequence similarity but have conserved structures [13] that consist of a six β-sheet sandwich containing two anti-parallel β-sheets, with both sheets formed by three β-strands (Figure 1A). All members have conserved β-strand orientations, with the first β-sheet formed by β1, β2, and β6, and the second formed by β3, β4, and β5, with a buried hydrophobic structural core. Most *M. oryzae* MAX effectors are recognised by rice heavy-metal-associated (HMA) domain-containing proteins [27,28], but they also target other proteins that may consist of multiple targets, such as in the case of AvrPiz-t [12,29,30,31]. The availability of these complexed structures allows for the benchmarking of effector–host receptor binding partner interactions. The binding interface of effector–plant protein complexes, interacting and non-interacting residues, and surface physico-chemical properties of both proteins can all be derived from the 3D structures of their complexes, which can be further applied in knowledge-based molecular docking. Once an optimised docking approach has been established based on all available experimental data, it can be applied in the docking of effectors, for which no 3D structure of their complexes with any plant protein is available. Ultimately, this information can also be used to screen for potential host receptor binding partners of effector proteins, and some effectors may be found to bind to different plant proteins that exhibit similar structures and protein domains.

To conduct this study, all available 3D structures of effector–plant protein complexes were identified, as well as the structures of the monomer (i.e., a single structure that was not obtained in a complexed form) of both effector proteins and plant proteins/receptors (Appendix A). All the available experimental structures of complexes belong to the MAX structural family, which include AvrPik variants, AvrPia, and Avr1CO39, all bound to heavy-metal-associated (HMA) domain proteins (Figure 1B). The structures contain hydrophobic surface patches that are putative sites for protein–protein interactions, and the binding of MAX effectors involves two distinct binding surfaces [28,33]. One interface involves the binding of effector AvrPik variants with the PikHMA protein, whilst the other interface involves the binding of effector Avr1CO39 with RGA5 [28]. Throughout this study, these two distinct binding surfaces are referred to as MAX regions 1 and 2. Most of the AvrPik-PikHMA interactions are formed with both the monomer or dimer of Pik, consisting of three major binding areas in the AvrPik-Pik complex: (a) a single hydrogen bond between the C-terminal β-strand of Pik and the β3 domain of AvrPik; (b) hydrogen bonds and salt bridges between the side chains of Asp224/225 in Pik and the side chains of Arg64 and Asp66 in AvrPik; and (c) the disordered N-terminal extension of polymorphic residues 46–48 in AvrPik forming hydrogen bonds/salt bridges and hydrophobic interactions with the C-terminal β4 in Pik [34]. The direct binding of Avr1CO39 to RGA5 occurs through a hydrophobic surface patch at the C-terminal region of RGA5, which is similar to that involved in the AvrPia-RGA5 interaction [28]. The same region is also involved in RGA5 homo-dimerisation, suggesting that binding to Avr1CO39 and AvrPia competes with RGA5 dimerisation [28]. Three binding areas are involved in the formation of the Avr1CO39-RGA5 complex, including residues Lys24, Asn37, Asn38, and Thr41, within the hydrophobic patch of Avr1CO39 [28]. A benchmarking analysis of the molecular docking approach was performed using these complex structures based on a ‘bound’ docking approximation, which involves the use of the 3D structures of each of the effector and plant proteins derived from their complex instead of their monomeric 3D structures. This allows the optimisation of the molecular docking protocol and the re-scoring of docking poses to reproduce the experimentally determined 3D structures of the complexes between effectors and plant proteins. After this, the so-called ‘unbound’ docking of the monomer structures of effector proteins and host receptor binding partners was conducted.

## 2. Results

The interaction between fungal effector proteins and their host receptor binding partners was investigated using molecular docking simulations in two parts. The first part (Section 2.1) was the benchmarking of the docking approach using experimentally determined complexed structures (bound docking) to optimise the method and identify the best docking parameters for this type of protein–protein complex. Benchmarking also determined whether ZDOCK and the optimised docking parameters can be successfully used in cases where information about the residues involved in the binding of effector–host receptor binding partners is not available. The second part (Section 2.2) involved the application of this approach and parameters to unbound structures (blind or unbound docking) in order to predict the interaction of fungal effector proteins and host receptor binding partners where no experimental complexed structures are available.

### 2.1. Benchmarking of Bound Complexes

ZDOCK was used in the initial stage search, resulting in 54,000 predicted docking poses per simulation. The pose with the highest ZDOCK score was initially deemed to be the best prediction. The following step determined whether the ZDOCK score alone could correctly rank near-native docking poses within the top-ranked binding poses. This benchmarking was performed using available 3D experimental structures consisting of eight effector–host receptor binding partner complexes: Avr1CO39-RGA5 [28], AvrPia-PikpHMA [35], AvrPikC-PikhHMA [36], AvrPikD-PikmHMA [14], AvrPikE-PikmHMA [14], AvrPikE-PikPHMA [14], AvrPikF-OsHIPP19 [37], and ApikL2F-sHMA94 [38] (Appendix A). All effectors and host receptors are from *M. oryzae* and *Oryza sativa*, respectively, except sHMA94, which is a receptor from *Setaria italica* [38]. Some of these complexes have several PDB entries, resulting in a total of 19 complexes, and each of these complexes was treated independently even though they correspond to the same oligomeric form. These experimental structures were used to validate the docking poses predicted using ZDOCK.

#### 2.1.1. Evaluation of Docking Output Using DockQ

The determination of near-native docking poses was completed by comparing docked poses with experimental structures as a reference using a quality assessment indicator, DockQ score. This is calculated based on interface RMSD (irms), ligand RMSD (lrms), energy score, and the fraction of common contacts (fnat) at the interface of the two interacting proteins [39]. The interacting or binding interface is defined as any pair of heavy atoms from the two molecules within 5.0 Å. Fnat is the fraction of native interfacial contacts preserved in the interface of the predicted docked complex compared to the experimental complexed structure. Ligand RMSD is the ligand RMSD calculated for the backbone of the shorter chain (effector protein/ligand) of the model after the superposition of the longer chain (host receptor binding partner/receptor). Interface RMSD is the receptor–ligand interface in the target (native) redefined at a relatively relaxed atomic contact cut-off distance of 10.0 Å, which is twice the value used to define inter-residue interface contacts in the case of Fnat. DockQ scores range from 0 to 1, with 1 being closest to the native structure. Docking poses with a DockQ score of above 0.5 were considered as acceptable, whilst scores above 0.8 were deemed to be good. Docking poses with the best DockQ score (closest to 1) for each bound case were evaluated and their position in the rank was recorded. DockQ scores were generated for each docked pose and the correspondence between the ZDOCK score and DockQ score was determined (Table 1) to evaluate the applicability of ZDOCK score for ranking near-native docking poses within the top predicted binding pose.

Table 1 reveals that the docking pose with the best ZDOCK score in 16 out of 19 complexes has a DockQ score above 0.8 (i.e., categorised as good). The docking poses with the best (highest) ZDOCK score for the remaining two complexes, Avr1CO39-RGA5 (PDB ID 5zngAC) and AvrPia-PikpHMA (PDB ID 6q76AB), have DockQ scores below 0.1, which are unreliable and indeed wrong, and we termed these complexes as non-ideal best poses (i.e., having low DockQ score). However, one complex, AvrPikE-PikpHMA (PDB ID 6g11EF), has a DockQ score of 0.753, which is still considered good. The top docking pose for two of the effector–host receptor binding partner complexes, AvrPikC-PikhHMA (PDB ID 7a8xEF) and ApikL2F-sHMA94 (PDB ID 7nmmDL), were ranked at the top by both ZDOCK and DockQ scores (Table 1, highlighted in bold), and we termed these complexes as ideal best pose (top-ranked ZDOCK DockQ scores). These two cases were the only ones in which the ZDOCK score was able to rank near-native poses at the top. However, it is reassuring that in most complexes the top-ranked docking pose on the basis of the ZDOCK score was close enough to the correct, near-native pose, as demonstrated by a high DockQ score.

It is important to contrast these predictions with those for complexes with non-ideal best pose, Avr1CO39-RGA5 (PDB ID 5zngAC) and AvrPia-PikpHMA (PDB ID 6q76AB), where the DockQ scores of their best ZDOCK score docking poses were 0.022 and 0.024, respectively, revealing that docking failed to predict the native/near-native binding pose. They also have the two lowest best ZDOCK scores of 46.35 and 43.46, contrasting with all other complexes with ZDOCK scores above 60. Interestingly, all other complexes involve interactions with the same binding site of plant HMA proteins, which we term as MAX region 1 throughout this study. In the case of Avr1CO39-RGA5 and AvrPia-PikpHMA, the effectors bind to a smaller region of plant HMA proteins, which we term as MAX region 2. Both of these distinct binding regions correspond to the protein surface regions identified by Guo et al. (2018) [28] and Cesari et al. (2022) [33]. It is also noteworthy that the docking search algorithm was indeed able to sample the correct (near-native) binding poses for these complexes, with the best DockQ scores being 0.967 for Avr1CO39-RGA5 and 0.962 for AvrPia-PikpHMA. However, as Table 1 shows, the corresponding ZDOCK scores were not the highest predicted. Avr1CO39-RGA5 and AvrPia-PikpHMA have different ZDOCK scores because the effectors (Avr1CO39 and AvrPia) interact with MAX region 2 compared to MAX region 1 for all other effectors described in this study. This differential interaction is probably due to the effector surface and polarity of the binding region, since all plant HMA domain proteins considered in this study have similar fold and surface physicochemical properties (regardless of the interaction with MAX region 1 or 2). Even though the Avr1CO39 and AvrPia effectors share a similar fold to other MAX effectors, both of them lack a 10-residue loop region in the N-terminal region, consisting of a combination of non-polar and polar residues, resulting in a lack of binding to MAX region 1. This loop region is present in the other effectors that bind to MAX region 1.

MAX regions 1 and 2 can be found on the surface of the same protein (Figure 1C), as is the case of plant PikpHMA. Two different fungal effectors can bind to the same HMA-containing protein utilising different (non-overlapping) binding regions (Figure 1B), and there is no evidence so far of simultaneous binding on both regions. Evidence of binding to MAX region 2 has only been reported for Avr1CO39 and AvrPia. Our docking simulations suggest that it may be necessary to optimise docking parameters for MAX region 1 and 2 independently.

We also examined the correlation of ZDOCK and DockQ scores for all 54,000 predicted docking poses in each bound case, as shown in Figure 2 and Appendix A. All scatterplots exhibit a funnel-shaped distribution towards high DockQ scores. There are some important differences in the shape of the scatterplots between complexes with ideal and non-ideal best poses. In complexes with an ideal best pose, a non-linear correlation is observed for docking poses with DockQ score values above 0.5 for AvrPikC-PikhHMA (PDB ID 7a8xEF) and ApikL2F-sHMA94 (PDB ID 7nmmDL) (Figure 2A,B). In complexes with non-ideal best poses, a more linear and flatter correlation is observed for all docking poses for Avr1CO39-RGA5 (PDB ID 5zngAC) and AvrPia-PikpHMA (PDB ID 6q76AB) (Figure 2C,D). This resulted in the near-native docking pose not being predicted to be within the top ZDOCK scores, such that this metric could not rank near-native docking poses highly enough.

These findings suggest that, for complexes with ideal best poses, and indeed the majority of bound complexes, the ZDOCK search method was able to identify and score highly enough near-native complexes, resulting in good DockQ scores (Table 1 and Appendix A). Figure 3A,B clearly shows how complexes with ideal best poses, AvrPikC-PikhHMA (PDB ID 7a8xEF) and ApikL2F-sHMA94 (PDB ID 7nmmDL), closely resemble near-native complex structures. By contrast, Figure 3C,D shows how complexes with non-ideal best poses, Avr1CO39-RGA5 (PDB ID 5zngAC) and AvrPia-PikpHMA (PDB ID 6q76AB), do not resemble the native structure of the complex at all and reflect the incorrect docking of the effector protein. In the case of Avr1CO39-RGA5 (PDB ID 5zngAC), the effector was incorrectly predicted to dock to the opposite binding region of the host receptor binding partner (Figure 3C), whilst in the case of AvrPia-PikpHMA (PDB ID 6q76AB), the effector was predicted to dock to the N-terminal region of the first β-strand, which is far away from the known MAX region 2 (Figure 3D). The lack of accuracy in the predicted binding poses is of course reflected by their very low DockQ scores.

In summary, the evaluation of the ZDOCK score using the DockQ score as a quality metric showed that the ZDOCK score can successfully rank near-native docking poses within the top 10 poses for most bound cases (Appendix A), completely failing to correctly predict the Avr1CO39-RGA5 (PDB ID 5zngAC) and AvrPia-PikpHMA (PDB ID 6q76AB) complexes. The latter appears to be related to the utilization of a different binding region (MAX region 2) on the HMA host receptor binding partner, whereas all other MAX complexes used for benchmarking involve binding to the larger region on the plant HMA protein (MAX region 1). Consequently, this study suggests that bound complexes involving MAX region 1 can be successfully predicted using the ZDOCK score, but not those involving MAX region 2. For the latter, the use of the ZDOCK score alone does not appear to be sufficient to successfully predict a near-native docking pose. In order to improve the ranking of near-native docking poses, especially in cases involving MAX region 2 (Avr1CO39-RGA5 and AvrPia-PikpHMA complexes), a subsequent stage of docking refinement was undertaken, which included the use of the ZRANK scoring function as well as additional scoring functions.

#### 2.1.2. Re-Scoring and Re-Ranking with ZRANK

By default, ZDOCK includes ZRANK scoring to re-score and re-rank the top 2000 ZDOCK score docking poses. This is the default cut-off since near-native structures have been reported to be within the top 2000 ZDOCK score poses [40]. ZRANK is a refinement step that includes a linear weighted sum of van der Waals attractive and repulsive energies, electrostatics short- and long-range attractive and repulsive energies, and desolvation [41]. Weights optimisation in ZRANK involve a smaller van der Waals repulsive term than the van der Waals attractive term, allowing for “softness” when evaluating rigid body predictions, such that docking poses with some degree of steric clashes can be ranked correctly. The repulsive electrostatics term is greater than the attractive term to help filter out predictions with unfavourable electrostatics as the repulsive force is deemed to be stronger than the attractive force. The long-range electrostatics term is larger than the short-range one. In this study, the top 2000 docking poses based on ZDOCK score for all bound complexes were assessed using ZRANK. The correspondence between the ZRANK score and the DockQ score is reported in Table 2.

Table 2 reveals that the docking pose with the best (lowest) ZRANK score in 14 out of 19 complexes has a DockQ score above 0.8 (i.e., categorised as good). The docking poses with the best ZRANK score for the other five complexes, AvrPia-PikpHMA (PDB ID 6q76AB), AvrPikD-PikmHMA (PDB ID 6fu9CD), AvrPikE-PikmHMA (PDB ID 6fubAB), and ApikL2F-sHMA94 (PDB ID 7nmmEM and 7nmmHP) all have DockQ scores below 0.25, which are unreliable and incorrect, and are also termed as complexes with non-ideal best poses. AvrPikC-PikhHMA (PDB ID 7a8xBC) and ApikL2F-sHMA94 (PDB ID 7nmmAI, 7nmmBJ and 7nmmFN) are complexes with ideal best poses (top-ranked by both ZRANK and DockQ scores) (Table 2, highlighted in bold). Unlike the ZDOCK score, the ZRANK score was unable to successfully rank a number of complexes that involve binding to the MAX region 1 as well as one involving binding to the MAX region 2.

We also examined the correlation of ZRANK and DockQ scores for all top 2000 docking poses in each bound case, as shown in Figure 4 and Appendix A. Unlike in the scatterplots of ZDOCK vs. DockQ score (Figure 2 and Appendix A), no clear funnel-shaped distribution can be observed. There were no differences observed in the shape of the scatterplots between complexes with ideal (Figure 4A,B) and non-ideal (Figure 4C,D) best poses. In general, a linear correlation is observed, which contrasts with the scatterplots for the equivalent correlation between ZDOCK and DockQ scores, which showed a non-linear distribution, especially for complexes with ideal best poses. As a consequence, docking poses ranked by ZRANK score with a DockQ score of above 0.5 cannot be discriminated as easily from those with DockQ scores below 0.5, limiting the applicability of using ZRANK scores for the ranking of docking poses.

The best examples of this limitation are not only complexes with best non-ideal pose ApikL2F-sHMA94 (PDB ID 7nmmHP and 7nmmEM) shown in Figure 4C,D, but also complexes AvrPia-PikpHMA (PDB ID 6q76AB), AvrPikA-PikmHMA (PDB ID 6fudAB), AvrPikD-PikmHMA (PDB ID 6fu9CD), and AvrPikE-PikmHMA (PDB ID 6fubAB), shown in Appendix A. In all of these complexes, the docking pose with the best ZRANK score pose has a very low DockQ score (<0.25), clearly revealing that these docking poses do not resemble the native complex structure at all.

The use of ZRANK scores for re-scoring and re-ranking the top 2000 ZDOCK docking poses appears to be inferior to the direct ranking with ZDOCK scores, even though some improvements in rank position of the best pose with ZRANK scores with respect to ZDOCK scores were observed. A comparison of Table 1 and Table 2 shows that, in general, the rank position of the best ZRANK score pose ranged from 1 to 522 compared to 1 to 1605 if based on the ZDOCK score.

#### 2.1.3. Comparison of Docking Predictions with ZDOCK and ZRANK Scores

To compare the use of ZDOCK and ZRANK scores to rank docking poses, we compiled a list of the best docking poses according to each scoring function as assessed by their DockQ scores, as shown in Table 3.

There were six complexes with significant changes in DockQ score (Table 3, highlighted in bold). Four complexes are now correctly predicted with the best ZDOCK score that were not with the best ZRANK score: AvrPikD-PikmHMA (PDB ID 6fu9CD), AvrPikE-PikmHMA (PDB ID 6fubAB), and ApikL2F-sHMA94 (PDB ID 7nmmEM, 7nmmHM). By contrast, there were two cases that were correctly predicted with the best ZRANK score but did not have the best ZDOCK score: Avr1CO39-RGA5 (PDB ID 5zngAC) and AvrPia-PikpHMA (PDB ID 6q76AB).

The use of ZRANK scores substantially worsened the ranking of near-native docking poses for four of the complexes (all involving MAX region 1): AvrPikD-PikmHMA (PDB ID 6fu9CD), AvrPikE-PikpHMA (PDB ID 6fubAB), ApikL2F-sHMA94 (PDB ID 7nmmEM), and ApikL2F-sHMA94 (PDB ID 7nmmHP). However, the use of ZRANK scores substantially improved the ranking of the near-native docking pose of Avr1CO39-RGA5 (PDB ID 5zngAC), which involved MAX region 2, as well as providing a relative improvement for the predicted near-native docking pose of AvrPikE-PikpHMA (PDB ID 6g11EF).

In summary, re-ranking using ZRANK score appears to worsen the ranking of near-native docking poses involving binding to MAX region 1 (such that ZDOCK scores should be preferred), but appears to improve the ranking of near-native docking poses involving binding to MAX region 2 to some extent. The availability of more experimentally determined complexes involving binding to MAX region 2 will allow for the better benchmarking of the best scoring function. Based on these findings, we proceeded to try to optimise docking parameters for both MAX regions 1 and 2 for use in unbound docking simulations in addition to a broader set of scoring functions (Section 2.2).

#### 2.1.4. Assessment of Docking Output with and without Residue Restraints

Before proceeding to further refine docking predictions using different scoring functions, the top 2000 ZDOCK score poses were assessed with and without the use of residue restraints. During benchmarking, the docking of bound complexes was also performed using residue restraints (through the use of passive and active residues) with the intent of assessing the ability of this docking approach to predict the native structures of the effector protein–host receptor binding partner complexes. In ZDOCK, passive residues were used during the initial docking stage as blocking residues to restrict the sampling search space, whilst active residues were implemented after docking to filter docking poses that satisfy all residues included. By contrast, the aim of docking without residue restraints (blind docking) was to determine the best docking approach and parameters for obtaining near-native docking poses that could be then used for the more challenging unbound docking. The use of restraints, however, does not seem to be practical in ZDOCK unless a limited number of restraints (fewer than five) are used, where more residues are preferable in the case of benchmarking. This is due to the very stringent requirement of using such restraints where all residues need to be satisfied (Appendix A, Appendix A).

### 2.2. Optimisation of Parameters for the Prediction of Near-Native Docking Poses

In unbound docking simulations, where the native structure of the effector–host receptor binding partner complex is not available, it is obviously not possible to use DockQ scores to identify the best docking poses. The previously described bound docking simulations suggest that there are three distinct scenarios: (a) binding to MAX region 1 in plant HMA proteins, (b) binding to MAX region 2 in plant HMA proteins, and (c) binding to non-MAX effectors (i.e., other effector families). The flat, near-linear correlation between ZDOCK and DockQ scores in the docking poses of effectors binding to MAX region 2 (Figure 2) suggests that the use of ZDOCK score to identify near-native poses is unreliable. However, it appears that, in general, the highest ZDOCK scores can be used to differentiate binding to MAX regions 1 and 2 in plant HMA proteins. For example, if ZDOCK outputs a docking pose with the highest ZDOCK score of above 50 then the predicted complex will likely bind to MAX region 1, and if the highest ZDOCK score is below 50 then the complex will likely bind to MAX region 2.

#### 2.2.1. Docking Parameters

This section discusses the docking parameters that were subsequently applied for unbound docking (Section 2.2.2) based on the benchmarking studies.

(a) MAX region 1 filter

The unbound docking of effectors and host receptor binding partners aimed at identifying a sufficiently good docking pose (i.e., DockQ score above 0.8) that would be deemed close enough to an ideal near-native pose (i.e., DockQ score close to 1). In order to find these docking poses, we created a filter termed ZDOCK50/ZRANK-80 which was specifically designed to retain near-native docking poses for effectors that bind to MAX region 1. This filter was adopted based on the correlation of ZDOCK versus DockQ scores of the benchmarking studies (Figure 2 and Appendix A). In the case of bound complexes involving MAX region 1, it was observed that the percentage of docking poses with a DockQ score above 0.5 was much higher than the percentage of poses with a DockQ score below 0.5 for poses with a ZDOCK score above 50. Consequently, the use of this filter for ZDOCK scores above 50 would likely eliminate unreliable docking poses (i.e., with DockQ score < 0.5). For some complexes, however, a significant percentage of unreliable docking poses still remained, leading to a subsequent filter being applied such that only docking poses with ZRANK score below −80 were retained, resulting in a minimal number of unreliable docking poses remaining (see Section 2.2.2). The resulting percentage of near-native poses was indeed higher compared to non-ideal best poses. For complexes where docking poses with a DockQ score below 0.5 still remained within the filtered pool, an application of a broader set of scoring functions derived from CCharPPI were included to eliminate those unwanted poses and improve the ranking of near-native poses. The docking poses of effectors interacting with MAX region 1 in plant HMA proteins and which had been processed using the above ZDOCK50/ZRANK-80 filter were re-scored using CCharPPI. The correlation between each one of the 108 scoring functions and the DockQ score was assessed for its ability to discriminate near-native from non-near-native docking poses.

The use of the ZDOCK50/ZRANK-80 filter resulted in 14 out of 17 complexes having poses with a DockQ score above 0.5, except for 5 complexes: AvrPikC-PikhHMA (PDB ID 7a8xEF) [36], AvrPikE-PikmHMA (PDB ID 6fubAB) [14], AvrPikD-PikmHMA (PDB ID 6fu9CD) [14], and ApikL2F-sHMA94 (PDB ID 7nmmBJ and 7nmmEM) [38]. The filtered docking poses for the above five complexes incorrectly predicted by the ZDOCK50/ZRANK-80 filter were re-scored using the scoring functions provided by CCharPPI with the aim of excluding docking poses with a DockQ score below 0.5. Scoring functions that were able to rank higher docking poses with DockQ scores above 0.5 were chosen. Among all scoring functions, only ELE was found to be able to filter out docking poses with DockQ scores below 0.5. ELE is the total electrostatic energy as calculated using PyDock, with a lower ELE score corresponding to a more favourable interaction [42].

Figure 5 shows the correlation between the ELE scoring function and DockQ scores. A cut-off of <−20 was set for the ELE scoring function, which resulted in poses with a DockQ score below 0.5 being removed, leading to the good discrimination of docking poses with DockQ scores above and below 0.5 for all five complexes, and providing good refinement of near-native docking poses after the use of the ZRANK50/ZRANK-80 filter. These filters and this re-scoring were chosen to be applied to the unbound docking of effectors to plant HMA proteins through MAX region 1.

(b) MAX region 2 filter

The MAX 2 region filter involves the application of the top 200 best ZRANK scores followed by refinement using 108 CCharPPI scoring functions. Out of 108 scoring functions, five were chosen as MAX region 2 filters since they were able to differentiate near-native and non-near-native rankings based on the benchmarking analysis performed using available MAX region 2 complexes, Avr1CO39-RGA5 (PDB ID 5zngAC) [28] and AvrPia-PikpHMA (PDB ID 6q76AB) [35]. The scoring functions selected were (1) CP_PIE, (2) AP_GOAP_ALL, (3) AP_OPUS_PSP, (4) CP_SKOa, and (5) CP_ZLOCAL_CB. CP_PIE is the PIE score [43], AP_GOAP_ALL is the total GOAP energy [44], AP_OPUS_PSP is the OPUS_PSP potential [45], CP_SKOa is the contact potential calculated between intermolecular residues [46,47], and CP_ZLOCAL_CB is the E_local Z-score C beta potential [48].

The correlations between the docking and DockQ scores shown in Figure 6 reveal that the use of the chosen CCharPPI scoring functions does not result in the adequate separation of near-native and non-near-native docking poses. However, the combination of these scoring functions resulted in the most optimised way of having near-native or docking poses with a DockQ score > 0.8 ranked as the top rank. The scoring functions have to be applied according to the following steps: (1) retain poses with CP_PIE > 0.8, (2) retain poses with AP_GOAP_ALL < 0, (3) retain poses with AP_OPUS_PSP < −100, (4) remove poses with CP_SKOa > 0, and (5) remove poses with CP_ZLOCAL_CB < 3. The application of these CCharPPI filters resulted in poses with DockQ scores above 0.8 only (Table 4). However, it should be acknowledged that the MAX region 2 filter might have limitations because it was benchmarked only using two experimental complexes representing MAX region 2 binding.

#### 2.2.2. Unbound Docking Results

In this section, we report the results of the unbound docking of MAX region 1 and 2. The unbound docking of effector proteins and host receptor binding partners made use of their structures experimentally resolved as monomers, or taken from the structure of a complex but docked to a different plant protein. Predicted models of effector proteins or host receptor binding partners for which there are no experimental structures were also included for unbound docking. Unbound docking was performed using ZDOCK using the parameters described in the previous section.

(a) Unbound docking predictions for effectors binding to HMA proteins through MAX region 1

The prediction of the binding of MAX effectors to plant HMA proteins involving MAX region 1 involved AvrPik variants (A, C, D, E, and F) and ApikL2F. Molecular docking was performed followed by the application of the ZDOCK50/ZRANK-80 filter to obtain the best (near-native) docking pose. The top poses obtained from the use of the ZDOCK50/ZRANK-80 filter are compiled in Table 5.

Upon the application of the ZDOCK50/ZRANK-80 filter, 44 out of 48 complexes satisfied the filter. The complexes that were excluded were AvrPikA-PikhHMA (PDB ID 6fubD-7a8x), AvrPikC-ancHMA (PDB ID 7a8xC-7bnt), AvrPikF-PikhHMA (PDB ID 7b1iC-7a8x), and ApikL2F-PikpHMA (PDB ID 7nmmI-5a6w). A total of 54% of the ranked poses were ranked as the top pose according to the ZRANK score, whilst 14.6% ranked as the top pose based on the ZDOCK score. Three complexes, AvrPikC- PikpHMA-mutant (PDB ID 7a8xC-6r8m), AvrPikC- ancHMA (PDB ID 7a8xC-7bnt), and AvrPikA-sHMA94 (PDB ID 6fubD-7nmm) had poses ranked as top pose based on both ZDOCK and ZRANK scores (Table 5, highlighted in bold).

Based on the predicted structures of the effector–host receptor binding partner complexes, all effectors bind to MAX region 1 in the plant HMA proteins. Interestingly, the complexes with the highest ZDOCK score only bind to two host receptor binding partners, osHIPP19 and PikpHMAmutant. Until now, no experimental structure of the complexes shown in Figure 7 has been reported, except for AvrPikF-osHIPP19 (PDB ID 7b1iBC) [37]. Our predictions suggest that these effectors may be potential binding partners of plant proteins osHIPP19 and PikpHMA. Furthermore, the validation of the unbound docking predictions was performed by computing the percentage of residues known to be involved in the interaction through MAX region 1 as well as in the effector protein (Table 6).

The best docked complexes were AvrPikC-PikpHMAmutant and ApikL2F-osHIPP19HMA, where all known interacting residues in both the effector and host receptor binding partner proteins were predicted at the interface. Most of the predicted unbound docking complexes utilised additional residues in the vicinity of known active residues, resulting in a higher number of interacting residues. An increase in the number of interacting residues between effector and host receptor binding partner proteins in the predicted docked complexes might also be influenced by the potential role of water molecules at the protein–protein interface and which would be replaced by amino acid residues, since this molecular docking study did not include the role of such water molecules. The interaction of residues with water molecules is indeed observed in all of the reference structures used.

(b) Unbound docking results of MAX region 2

The prediction of the binding of MAX effectors to plant HMA proteins involving MAX region 2 involved two complexes, AvrPia (PDB ID 6q76B) with RGA5 (PDB ID 5zngA), and Avr1CO39 (PDB ID 5zngC) with PikpHMA (PDB ID 6q76A). Molecular docking was performed followed by the application of the CCharPPI filter to obtain the best (near-native) docking pose. Upon application of the CCharPPI filter, both complexes satisfied the filter, and the top docking poses are displayed in Figure 8.

Based on the predicted structures of the effector–host receptor binding partner complexes, all effectors bind to MAX region 2 in the plant HMA proteins. Currently, there are no experimental structures available for the complexes shown in Figure 8. Our prediction supports the finding of the recognition of AvrPia by RGA5 through direct interaction [11,27,28] (Figure 8A), and suggests that effector Avr1CO39 may be a potential binding partner of PikpHMA (Figure 8B). Furthermore, validation of the unbound docking predictions was done by computing the percentage of residues known to be involved in the interaction through MAX region 2 as well as in the effector protein based on the experimental complexes of Avr1CO39-RGA5 (PDB ID 5zngAC) [28] and AvrPia-PikpHMA (PDB ID 6q76AB) [35]. Both unbound docking complexes predicted utilised residues in the vicinity of known active residues except for residues on the RGA5 plant protein (Table 7), which is also known to be involved in the self-interaction of RGA5 [28].

As for the prediction of the binding of effectors to plant HMA proteins involving MAX region 2, two MAX effector candidates whose structure had been previously predicted through template-based modelling using RaptorX were considered: M.BR29.EuGene_00106461 and phenotypically validated effector SPD7 [32]. Structural modelling allowed SPD7 to be predicted to belong to the MAX structural family. Both MAX effectors were docked to the PikpHMA (PDB ID 6q76A) protein, which was specifically selected because different fungal effectors have been known to bind to both MAX regions 1 and 2 of PikpHMA. Unbound docking was thus used to assess the potential of SPD7 to bind to HMA plant proteins and its preference for either MAX region 1 or 2 on the basis of the magnitude of the best ZDOCK score: if this was above 50 then SPD7 would be deemed to bind to MAX region 1, and if it was below 50 it would be deemed to bind to MAX region 2. M.BR29.EuGene_00106461 and SPD7 were predicted to bind with top ZDOCK scores of 41.74 and 49.9, respectively. This suggests that both of these MAX effector proteins bind to MAX region 2 of PikpHMA. This is indeed consistent with the effector templates used to predict their structures, PDB structures 2mywA and 5zngC for M.BR29.EuGene_00106461 and SPD7, respectively, which involve binding to MAX region 2. Unfortunately, the application of the CCharPPI filter to MAX region 2 unbound docking poses was unsuccessful since there was no pose that passed the second step of the filter. This reflects the substantial limitation of benchmarking using a very small number of experimental complexes.

## 3. Discussion

### 3.1. Interactions with MAX Regions 1 and 2

Plant HMA domain-containing protein PikpHMA possesses two surface regions for effector protein binding, which were termed here as MAX regions 1 and 2. Plant PikpHMA was characterised in experimentally determined complexed structures with AvrPikh (PDB ID 6q11) bound at MAX region 1 and AvrPia (PDB ID 6q76) bound at MAX region 2. These experimental structures reveal the presence of 14 hydrogen bonds and 10 salt bridges in the interaction with MAX region 1, and 9 hydrogen bonds and 2 salt bridges in the interaction with MAX region 2. From the perspective of PikpHMA, the number of interfacial residues involved in the interaction in MAX regions 1 and 2 are 27 and 14, respectively, with 102 and 53 atoms found at the interacting interface in MAX regions 1 and 2, respectively. The solvent-accessible surface areas of MAX regions 1 and 2 are 967.6 and 475.9 Å, respectively, making MAX region 1 larger than region 2. The surface of MAX region 1 has a combination of polar and hydrophobic residues, whilst MAX region 2 only has hydrophobic residues. Differences in the polarity of MAX regions 1 and 2 resulted in the variation of ZDOCK and ZRANK scores for these two regions.

From the perspective of the fungal effector proteins that bind to these MAX regions on plant HMA domain proteins, the number of residues at the interacting interface for MAX regions 1 and 2 are 32 and 12, respectively, with 96 and 47 atoms at their interacting interfaces with MAX regions 1 and 2, respectively. The solvent-accessible surface areas of the effector interfaces for MAX regions 1 and 2 are 913.5 and 445.5 Å, respectively. The interacting surface of the effector with MAX region 1 is larger than the one with region 2, and the percentage of interacting residues was higher in the effector binding to MAX region 1 compared to the plant interacting surface in MAX region 1. By contrast, the plant interacting surface in MAX region 2 has a higher percentage of interacting residues compared to the interacting residues in the effector protein that binds to it. The solvent-accessible surface area was higher in MAX regions 1 and 2 compared to the effector interfaces.

The detailed analysis of the interactions involving MAX regions 1 and 2 described in this study had not been previously reported. This was necessary to further distinguish the alternative binding of MAX effector proteins to their plant partner proteins through these two different binding regions, for which different docking approaches appear to be necessary as they should be considered to be independent binding sites. Similar approaches could be applied to molecular docking simulations of effector–host receptor binding partner interactions involving other fungal effector structural families, but we cautiously speculate that each family may require further optimisation. The benchmarking studies reported in this study will likely only be applicable to the MAX effector family and the reported optimal docking parameters would require modification if applied to other effector families.

### 3.2. Polymorphic Residues and Engineered Plant HMA Domain Proteins

This study included an engineered PikpHMA protein (PDB ID 6r8m) that had the mutations Asn261Lys and Lys262Glu [49] and which involve MAX region 1. Asp261 is also a polymorphic residue of PikhHMA. The presence of polymorphic residues on the surface of PikpHMA and PikhHMA affect their interactions with fungal effector proteins. In plants in general, the presence of polymorphic residues indicates the occurrence of an evolutionary arms race in plant immune-related proteins aimed at combating the actions of fungal effectors, leading to heightened plant protein sensitivity towards effector proteins. Consequently, engineering these sites can play an important role in improving the receptor efficiency of binding and interaction with effector proteins in general [49].

Most of the polymorphic residues have been found in MAX region 1 but not region 2, and we can only discuss MAX region 1 in the case of the engineered plant protein since mutations are only present there. Compared to the non-mutated PikpHMA, mutations Asn261Lys and Lys262Glu involve major changes to the charge and size of the residues, resulting in an increase in the total number of hydrogen bonds at the interacting interface between effector and plant protein from 14 to 16, and the same total of 10 salt bridges. From the perspective of the plant protein, the percentage of residues involved at the interacting interface in MAX region 1 in the non-mutated (6g11BC) and mutated (6r8mFG) forms are 27 (37%) and 28 (36.4%), respectively, with 102 and 97 atoms at the interface, respectively. The solvent-accessible surface area of MAX region 1 in the non-mutated form is 967.6 Å and 988.3 Å in the mutated form.

Docking simulations revealed a higher ZDOCK score and better rank for the near-native pose for mutated PikpHMA compared to non-mutated PikpHMA. This might be due to the larger interacting surface area of mutated PikpHMA compared to its non-mutated form. It has also been shown that the mutation of Asn261Lys increases binding affinity to effectors [36]. Mutations also affect surface polarity, such that mutation Asn261Lys led to the formation of hydrogen bonds between Ser72 and Glu53 with Lys261, as well as a salt bridge with Glu53. At the same time, the Asn261Lys mutation also disrupts hydrogen bonds and salt bridges previously present in non-mutated PikpHMA at position Lys262. These mutations in MAX region 1 of PikpHMA might also make the engineered form more favourable for binding to MAX effectors, and this can indeed be observed in the docking predictions for unbound complexes, where two MAX effectors have better ZDOCK scores when they bind to the mutated form of PikpHMA instead of the non-mutated form, whilst the remaining four effectors bind to plant protein osHIPP19. The predicted complex with the engineered PikpHMA form was predicted to have the highest ZDOCK scores.

A good docking prediction of the formation of a complex between a MAX effector and a plant HMA domain protein may also be able to distinguish between effector protein binding regions and plant protein binding regions (i.e., dimerisation sites). At the same time, docking predictions might mistake dimerisation sites observed in some of the experimental complex structures (e.g., PDB structures 5r8m and 6g11) for the favoured interaction site, since it may fulfil the requirement of a protein–protein interaction region during the initial stage of docking (i.e., sampling), or in the prediction of potential protein–protein interaction binding sites.

### 3.3. Best Scoring Function for MAX Regions 1 and 2

It is interesting to discuss the reasons why the ZDOCK score was found to be better at ranking near-native docking poses amongst the top poses for MAX region 1 but not for MAX region 2. As summarised in the Methods section, ZDOCK scoring has three components (pairwise shape complementarity (PSC), desolvation (ACE), and electrostatics energy terms) and the docking pose with the highest ZDOCK score thus has the best combination of pairwise shape complementarity, desolvation, and electrostatics. The influence of each term can manifest in the specific docking predictions made for each MAX binding region.

MAX region 1 has a higher number of atom pairs between the effector and host receptor binding partner, which results in higher PSC scores compared to region 2. PSC is commonly used as a scoring function and, in this study, it was good at distinguishing near-native docking poses based on the interface features of protein complexes for MAX region 1 but not for region 2. The inability of PSC to describe the correct interaction in MAX region 2 might be due to its flatter protein–protein interface, and the ZDOCK score is known to have a preference for identifying large concave binding pockets [50]. The PSC function in ZDOCK is effective at discriminating large pockets in the binding sites of enzymes or inhibitor complexes, but has not been reported on its application in the case of effector–host receptor binding partner protein complexes.

MAX region 1 has a higher amount of protein–water atom contacts compared to MAX region 2, with 17 water oxygen atoms in MAX region 1 and 7 water atoms in MAX region 2, resulting in a higher total desolvation score for MAX region 1 compared to region 2. The larger interface binding area in MAX region 1 results in the burial of a larger surface area, which contributes to a stronger van der Waals contact energy in MAX region 1 compared to region 2. However, the downside of relying solely on van der Waals (vdW) interactions is their high sensitivity to incorrect side chain rotamers. Therefore, it is better to decrease the weight of vdW energies. The electrostatic potential of the plant HMA domain protein is determined by charged and polar residues, and there are four charged and four polar residues in MAX region 1, and one charged and three polar residues in MAX region 2. The higher scores of PSC, ACE, and electrostatics for MAX region 1 led to the success of the ZDOCK score when ranking near-native docking poses among the top poses, and justifies why the ZDOCK score does not work for binding to MAX region 2, even though both regions are on the surface of the same plant protein. There might also be a bias in the electrostatics scoring function of ZDOCK since its benchmarking was performed with an antibody–antigen complex test case [50]. It is clear that similar issues might apply to other potential host receptor binding partners with multiple effector binding surfaces.

The inability of the ZDOCK score to work effectively with MAX region 2 could also be due to the poor binding affinity of the docked poses and large backbone conformational changes in that region [50]. Protein surface side chains may need to undergo conformational changes upon binding. So far, there have been no reports of conformational changes upon binding for effector–host receptor binding partner interactions, especially for MAX region 2, but such occurrence could limit the success of docking during its initial sampling stage. However, the benchmarking reported in this study with 54,000 docking poses showed that the best DockQ score for both MAX regions 1 and 2 was above 0.9, indicating that near-native poses exist and sampling was successful, but the binding poses could not be ranked correctly in the case of MAX region 2. Sampling issues might be more applicable in the case of unbound docking, where conformational changes are more likely to be necessary for optimal binding, and, of course, this cannot be simply addressed with the use of a scoring function more tolerant to conformational changes. Rigid body docking might be able to address small conformational changes upon binding, but larger conformational changes will likely require flexible docking approaches.

The general benchmarking of ZRANK has shown improvements in the success rate of the top rank over the ZDOCK rank, but not in this study. The ZRANK score is meant to be used for the refinement step after ranking with ZDOCK. Based on our findings, the ZRANK score does not perform well for both MAX regions 1 and 2. Whilst it might slightly improve the rankings of docking poses involving MAX region 2, overall, the ZRANK score was not shown to be good at ranking near-native or good-enough poses as the top pose. The ZRANK score is based on the linear weighted sum of van der Waals attractive and repulsive energies, electrostatic short- and long-range attractive and repulsive energies, and desolvation [41]. The optimisation of weights in ZRANK involves a smaller van der Waals repulsive term than the van der Waals attractive term, allowing for “softness” when evaluating rigid body predictions, such that docking poses with some degree of steric clashes can be ranked correctly. The repulsive electrostatic term is greater than the attractive term to help filter out predictions with unfavourable electrostatics, as the repulsive force is stronger than the attractive force and the long-range electrostatic term is larger than the short-range one. One main difference between the ZDOCK and ZRANK scoring functions is the absence of pairwise shape complementarity in the latter. The binding interfaces of the MAX regions are probably best defined by their shape complementarity compared to optimised weighted van der Waals and electrostatic energies. Similarly to the ZDOCK score, the ZRANK score is also limited to conformations that do not change upon binding [41].

### 3.4. Limitations of Molecular Docking and Future Improvements

Based on our findings, the docking of effectors involving MAX region 2 is a case that requires more investigation since only two experimentally determined complexes are available, which imposes limitations for the proper benchmarking of the docking approach. Similar challenges may arise for the docking of fungal effectors belonging to other structural families that also lack experimentally determined complexes. For example, it would be challenging to investigate unbound docking involving the ToxA-like effector family because our benchmarking was specific for MAX complexes, and even within MAX complexes, different binding sites require different optimised docking parameters, as is the case for MAX regions 1 and 2. This problem will be overcome in future when more experimentally determined complexes of effector–host receptor binding partners are obtained. Docking improvements that can be implemented in the future for predicting the interactions of effector–host receptor binding partner include the use of flexible molecular docking, such as that implemented in HADDOCK [23,51], which allows for the option of including water molecules and allows conformational flexibility during docking at the atomistic level. It may also address the issue of complementary shape upon binding [52].

Another option is to apply template-based docking, which uses available experimentally determined complex structures as templates during docking, for example, by using the InterEvDock3 tool [53]. Since most of the available experimentally determined structures were based on MAX complexes, a template-based docking approach could be applied to study protein–protein interactions involving MAX effectors with plant HMA domain proteins. This approach could also be used to screen for potential effectors and plant proteins with similar structures. Alternatively, a broader set of scoring functions to better model the different aspects of effector–host receptor binding partner interactions could be applied, including the application of machine-learning-based scoring functions such as MetaScore, which have shown improvements over traditional scoring functions [54].

## 4. Materials and Methods

### 4.1. Experimental 3D Structures

For the benchmarking of molecular docking simulations, only 3D structures of effector proteins in complex (bound) with their host receptor binding partners were downloaded (in .pdb format) from the RSCB Protein Data Bank (PDB), as summarised in Appendix A. The 3D monomer structures (unbound) of effector proteins and host receptor binding partners were also downloaded from the RSCB PDB. In the benchmarking studies, the experimental 3D structures of effector–host receptor binding partner complexes were assessed using PyMOL Open Source (Schrödinger, LLC, New York, NY, USA) to determine the interactions between proteins and produce a list of interacting residues. Each complexed structure was then separated to produce separate PDB files of the effector protein and host receptor binding partner. These structures were pre-processed including energy minimisation (see further below) before proceeding with molecular docking using ZDOCK. Residues identified in the assessment of interactions present in the complexed structure were categorised as active or passive residues restraints. Active residues are residues on the surface of the protein determined to be directly involved in the interaction between the effector protein and host receptor binding partner, whilst passive residues are residues that are known not to be involved in the interaction and, usually, they are determined from Y2H assays that show negative results for the two interacting residues. In this study, only active residues were included as docking parameters to assist in the sampling stage, where the rotational and translational sampling of one protein (the effector protein) was conducted with respect to another stationary protein, which in this case was the host receptor binding partner.

### 4.2. Protein Structure Modelling

In the case of ‘unbound’ docking (see further below), the 3D structures of fungal effector proteins were modelled using both template- and non-template-based modelling approaches reported recently [32,55]. As for the interacting host receptor binding partners, template-based modelling was applied using SWISSMODEL, ITASSER, and RaptorX, whilst non-template-based modelling was implemented using QUARK, Robetta, and Rosetta. The best final model from each method was selected for unbound docking.

### 4.3. Protein Structure Processing

All PDB structures were prepared with the Prepare Protein option in Biovia Discovery Studio version 2020 (Dassault Systèmes, San Diego, CA, USA) using default parameters. The structures were checked for missing atoms, atom names, alternate conformations, incomplete residues, and bond orders. Missing internal loop regions were filled, hydrogen atoms were added, and the final structure was protonated by calculating protein ionisation and residue pKas. Energy minimisation was conducted for each structure using a maximum of 200 steps, and including the Generalized Born implicit solvent model. In the case of ‘bound’ complexes, each one of the proteins was separated before energy minimisation. Interacting residues at the interface of effector protein and host receptor binding partner complexes were identified using PyMOL Open Source (Schrödinger, LLC, New York, NY, USA) with the plugins interfaceResidues.pl and listcontacts.pl (https://pymolwiki.org/index.php/InterfaceResidues, accessed on 1 February 2021). A cut-off distance of 4.0 Å was used to identify the interacting residues. For ‘bound’ structures, the Euler coordinates of the ligand were randomised using PyMOL prior to molecular docking.

### 4.4. Protein–Protein Docking

In general, protein–protein docking was performed in two main stages: initial search and refinement. In the initial stage search, otherwise known as sampling, a large number of possible docking (binding) poses were predicted using the docking program. During the subsequent refinement stage, a small number of top-ranked docking poses from the initial stage search were refined and re-ranked using a more detailed scoring function.

Rigid docking was conducted using ZDOCK version 3.0.2 [40]. ZDOCK is a rigid docking program that uses 3D grid-based spatial searches during the sampling of binding poses using an efficient fast Fourier transform (FFT) method [40]. In rigid body docking, the bond angles, bond lengths, and torsion angles of both interacting proteins are kept fixed throughout the docking process. ZDOCK performs an exhaustive rigid body search in the 6D rotational and translational space, resulting in a large number of predicted docking poses. Each one of these poses was assigned a ZDOCK score, composed of pairwise shape complementarity (PSC), desolvation, and electrostatics energy terms. PSC computes the total number of atom pairs between the two proteins being docked within a distance cut-off, and a penalty term is assigned to grid point overlap of core–core, surface–core and surface–surface terms to prevent steric clashes [40,56]. Desolvation is based on the atomic contact energy (ACE), which is the free energy change of breaking two protein atom–water contacts and replacing them with a protein–atom–protein atom contact and a water–water contact. The total desolvation score was the sum of the ACE scores of all receptor–ligand atom pairs within a distance cut-off of 6.0 Å. Electrostatics is calculated using Coulomb’s law equation, which is calculated as a function of the electrostatic potential generated by the receptor and the partial charges of ligand atoms. The docking pose with the highest ZDOCK score was considered to have the best combination of PSC, desolvation, and electrostatics energies.

ZDOCK simulations were undertaken using Biovia Discovery Studio version 2020 (Dassault Systèmes, San Diego, CA, USA). The default settings had an angular step size of 6, generating 54,000 docking poses. Each score component of ZRANK was reported and advanced settings for filtering initial sampling poses using electrostatics and desolvation energies were selected. No blocked residues, or residues known not to form any interaction (‘passive residues’), were included in both fungal effector protein and host receptor binding partners. Docking poses were filtered using binding site residues or other residues known to form interactions (‘active residues’) in both the receptor and ligand (Appendix A). The binding interface of the receptor was defined as the set of receptor residues that had atoms within 10.0 Å of at least one ligand atom, and the same was achieved with the ligand-binding interface, which is the set of ligand residues with at least one atom within 10.0 Å of any receptor atom. Only heavy atoms were considered. A docking pose was required to have all active residues listed on both the receptor and ligand to pass filtering. The top 2000 poses ranked based on the ZDOCK score were re-ranked using ZRANK, which has more detailed electrostatics, van der Waals, and desolvation energy terms [41].

In the benchmarking of ‘bound’ complexes, effector proteins were docked to their corresponding host receptor binding partner followed by the filtering of the initial docking poses using two different modes: (1) with residue restraints, which are the active residues on both the receptor and ligand, and (2) without residue restraints (blind docking). A comparison of the docking poses to the native bound complexes was performed using DockQ [39], which provides a DockQ score between 0 and 1 (with 1 being closest to the native structure) based on the fraction of native contacts (fnat), interface RMSD (iRMSD), and ligand RMSD (lRMSD) [39]. Docking poses with a DockQ score above 0.5 were considered to be acceptable, whilst scores above 0.8 were deemed to be good. The docking pose with the best DockQ score (closest to 1) for each ‘bound’ case was evaluated by comparing it with a native reference complex, and its position in the rank was recorded.

In the benchmarking of ‘unbound’ complexes, there were three simulation categories depending on the type of unbound structures used: (1) the docking of effector and host receptor binding partners with 3D experimentally determined structures for which (1a) ‘monomer’ structures were taken from the complexed structure of effector and plant proteins, which were then docked to the effector or host receptor binding partner that originated from a different complex, or (1b) both structures were determined in their monomeric form; (2) the docking of either effector or host receptor binding partner with a 3D experimentally determined structure, and the other being a modelled structure; and (3) the docking of the predicted models of both effector and host receptor binding partners.

### 4.5. Re-Scoring and Re-Ranking

A subset of docking poses were re-ranked using CCharPPI (https://life.bsc.es/pid/ccharppi, accessed on 5 May 2021) [57]. CCharPPI is a webserver that compiles 108 scoring functions or descriptors from different sources related to molecular docking and protein–protein interactions. These scoring functions were categorised based on their general function into 10 different categories: atomic contact/step potentials, statistical potential constituent terms, van der Waals and electrostatics, solvation energy functions, residue contact/step potentials, residue distance-dependent potentials, atomic distance-dependent potentials, hydrogen bonding composite scoring functions, and miscellaneous [41]. Among these scoring functions, the best scoring functions that could identify the closest pose to the native structure were selected.

## 5. Conclusions

The characterisation of protein–protein interactions between effectors and their host receptor binding partners is crucial for understanding infection mechanisms in important agricultural crops, which may lead to the development of breeding for resistance cultivars in cases where there are direct interactions between the effectors and plant-resistant proteins. The formation of effector–host receptor binding partner complexes is the hallmark of the survival of both species. These interactions are usually transient, occurring only during disease infections and returning to a normal state once their objective has been achieved. Changes in the protein interfaces and compatibility that occurs following effector–host receptor binding partner interactions leads to disease infections.

We used ZDOCK to undertake ‘bound’ and ‘unbound’ docking simulations of the interactions of MAX effector proteins with their plant HMA domain proteins, as well as to characterise the predicted interactions. We identified major differences in the interactions relating to the presence of two different binding sites in plant HMA domain-containing proteins, which we termed MAX regions 1 and 2. We optimised docking parameters to rank docking poses involving these binding regions, as they were shown to exhibit major differences in binding due to different polarities at the interacting sites, necessitating two different docking approaches. This allowed for the use of blind docking, where no residues were used as restraints.

This study was able to differentiate effectors that bind to different regions on plant HMA domain proteins. Based on the unbound docking simulations, this approach missed 5–13.4% of the interacting residues involved in MAX binding region 1 for 4 out of 12 complexes. The limited number of complexes prevents us from making similar assertions about MAX binding regions 2. On the basis of the optimised docking parameters that were applied to unbound docking, this approach seems to be reliable enough to investigate MAX effector candidates that bind to plant HMA proteins involving MAX binding region 1. More optimisation and work will be needed for MAX region 2.

Benchmarking was only conducted with MAX effectors. More studies are thus needed to apply a similar benchmarking approach to effector–host receptor binding partner complexes outside of the MAX structural family. This will be facilitated when more experimentally determined structures of effector–host receptor binding partner complexes become available, especially for non-MAX effector families. Improvements could also be made by using better scoring functions for ranking docked poses, particularly for the initial stage of docking.

Improvements in these computational approaches will contribute to the future discovery of more Avr-R/NE-S interactions upon experimental validation. The approaches proposed here could also be applied to effector/target interaction and allelic variants of these targets. Understanding the interaction of fungal effector proteins with their host receptor binding partners will provide important information on their mechanisms of action and may contribute to expanding the database of possible effector—R/S-gene interactions in the effector-assisted breeding of new cultivars with disease resistance [15].

## Figures and Tables

**Figure 1 ijms-24-15239-f001:**
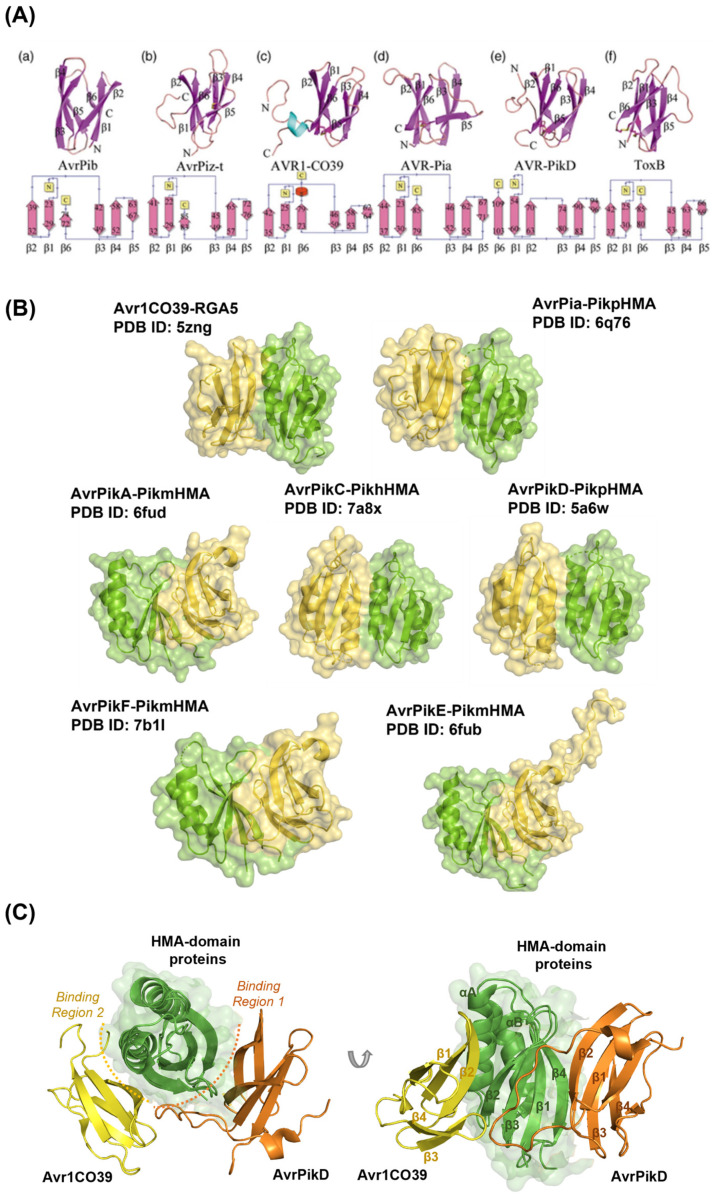
(**A**) Structures of effector proteins from the MAX family: (a) AvrPib, (b) AvrPiz-t, (c) Avr1CO39, (d) AvrPia, (e) AvrPikD, and (f) ToxB. The β-strands are shown in purple, and the α-helix is shown in cyan. The structural topology of the secondary structures is shown below each protein. Panel (**A**) was adapted from Rozano et al. (2023) under the Creative Commons CC-BY 4.0 license [32]. (**B**) Structures of effector proteins Avr1CO39, AvrPia, and AvrPik variants (yellow) in complex with their host receptor binding partners (heavy-metal-associated (HMA) domain proteins) (green). Structures are shown in combined ribbon and surface representation. The PDB ID of each complex is included. (**C**) Binding interfaces of HMA domain proteins visualised from above (**left** panel) and front (**right** panel) consisting of Avr1CO39-RGA5 (in yellow) and AvrPikD-PikpHMA (in orange) shown in combined ribbon and surface representation. Binding interfaces are represented as a dotted line in orange and yellow, representing the first and second interface, respectively. Throughout this study, the first interface (in orange) is referred to as MAX region 1, and the second interface (in yellow) as MAX region 2.

**Figure 2 ijms-24-15239-f002:**
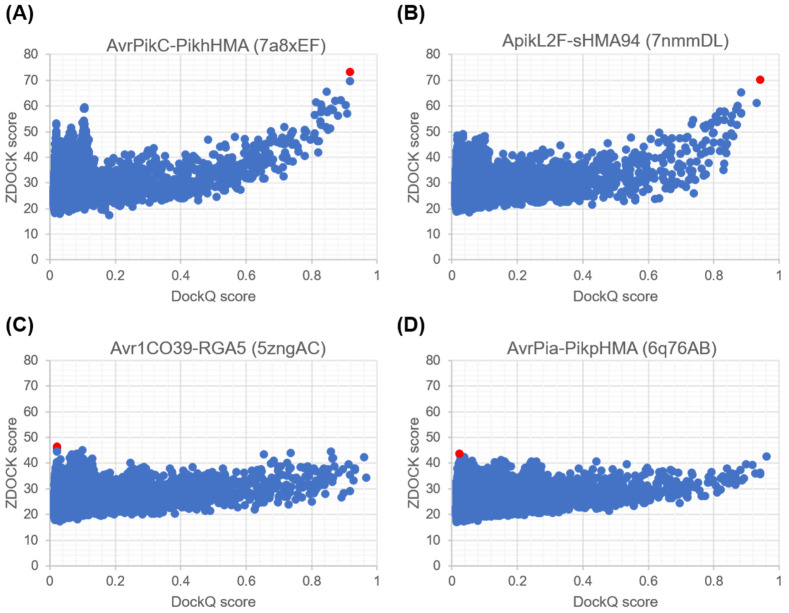
Correlation between ZDOCK and DockQ scores for all 54,000 predicted docking poses. The plots shown correspond to effector–host receptor binding partner complexes with ideal best poses, (**A**) AvrPikC-PikhHMA (PDB ID 7a8xEF) and (**B**) ApikL2F-sHMA94 (PDB ID 7nmmDL), as well as complexes with non-ideal best poses, (**C**) Avr1CO39-RGA5 (PDB ID 5zngC) and (**D**) AvrPia-PikpHMA (PDB ID 6q76AB). The docking pose with the best ZDOCK score is shown in red in each plot. Plots for the remaining bound complexes used in the benchmarking of ZDOCK scores are reported in Appendix A.

**Figure 3 ijms-24-15239-f003:**
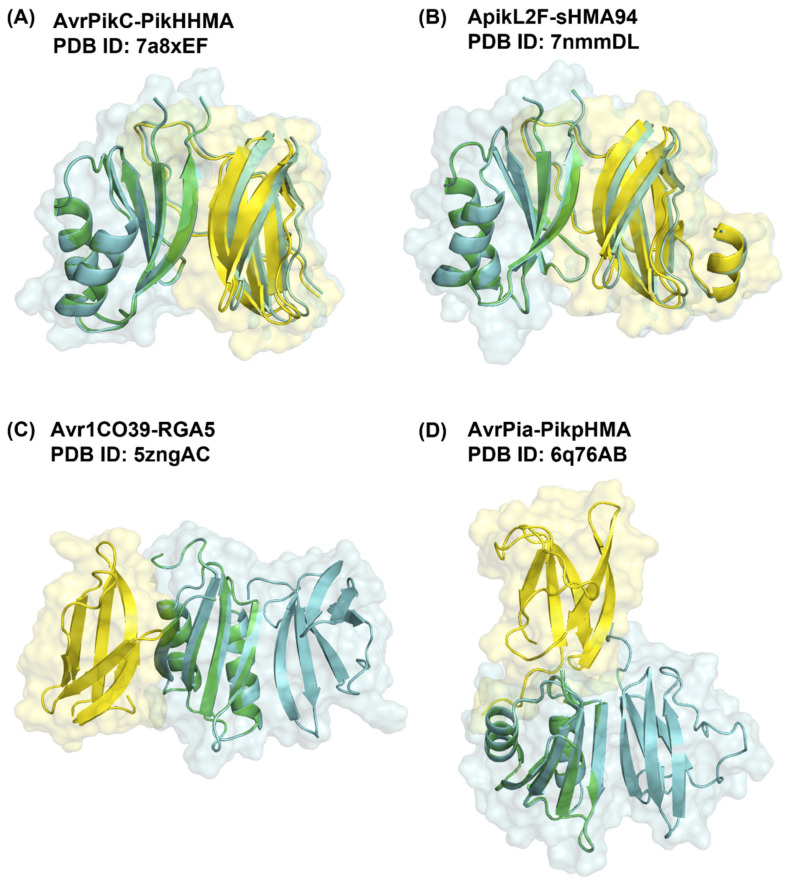
Predicted 3D structure of complexes with ideal best poses, (**A**) AvrPikC-PikhHMA (PDB ID 7a8xEF) and (**B**) ApikL2F-sHMA94 (PDB ID 7nmmDL), and complexes with non-ideal best poses, (**C**) Avr1CO39-RGA5 (PDB ID 5zngAC) and (**D**) AvrPia-PikpHMA (PDB ID 6q76AB), superimposed onto their respective reference structure. The predicted docking pose of the effector and host receptor binding partner is shown in yellow and green, respectively, with the reference complex shown in cyan. All structures are shown in ribbon representation with transparent surface representation for the reference complex (light cyan) and the predicted docked effector (light yellow).

**Figure 4 ijms-24-15239-f004:**
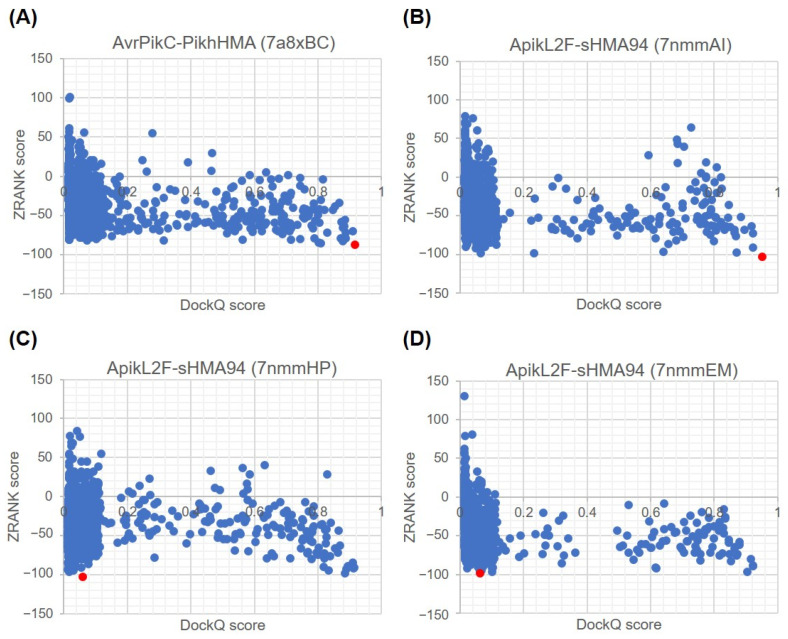
Correlation between ZRANK and DockQ scores for the top 2000 predicted docking poses using the ZDOCK score. The plots shown correspond to effector–host receptor binding partner complexes with ideal best poses, (**A**) AvrPikC-PikhHMA (PDB ID 7a8xBC) and (**B**) ApikL2F-sHMA94 (PDB ID 7nmmAI), as well as complexes with non-ideal best poses, (**C**) ApikL2F-sHMA94 (PDB ID 7nmmHP) and (**D**) ApikL2F-sHMA94 (PDB ID 7nmmEM). The docking pose with the best ZRANK score is shown in red in each plot. Plots for the remaining bound complexes used in the benchmarking of ZRANK scores are reported in Appendix A.

**Figure 5 ijms-24-15239-f005:**
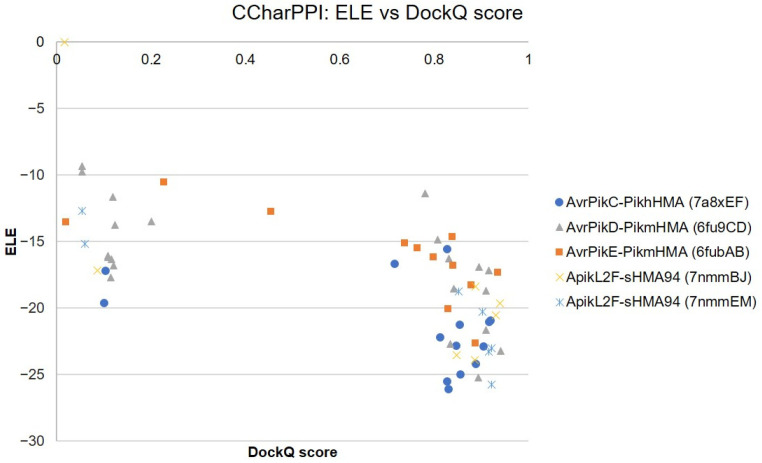
Correlation between ELE scoring function score and DockQ score for complexes AvrPikC-PikhHMA (PDB ID 7a8xEF) as blue round circle, AvrPikE-PikmHMA (PDB ID 6fubAB) as orange square, AvrPikD-PikmHMA (PDB ID 6fu9CD) as grey triangle, ApikL2F-sHMA94 (PDB ID 7nmmBJ) as yellow cross, and ApikL2F-sHMA94 (PDB ID 7nmmEM) as blue cross. Plots for all other bound complexes are reported in Appendix A.

**Figure 6 ijms-24-15239-f006:**
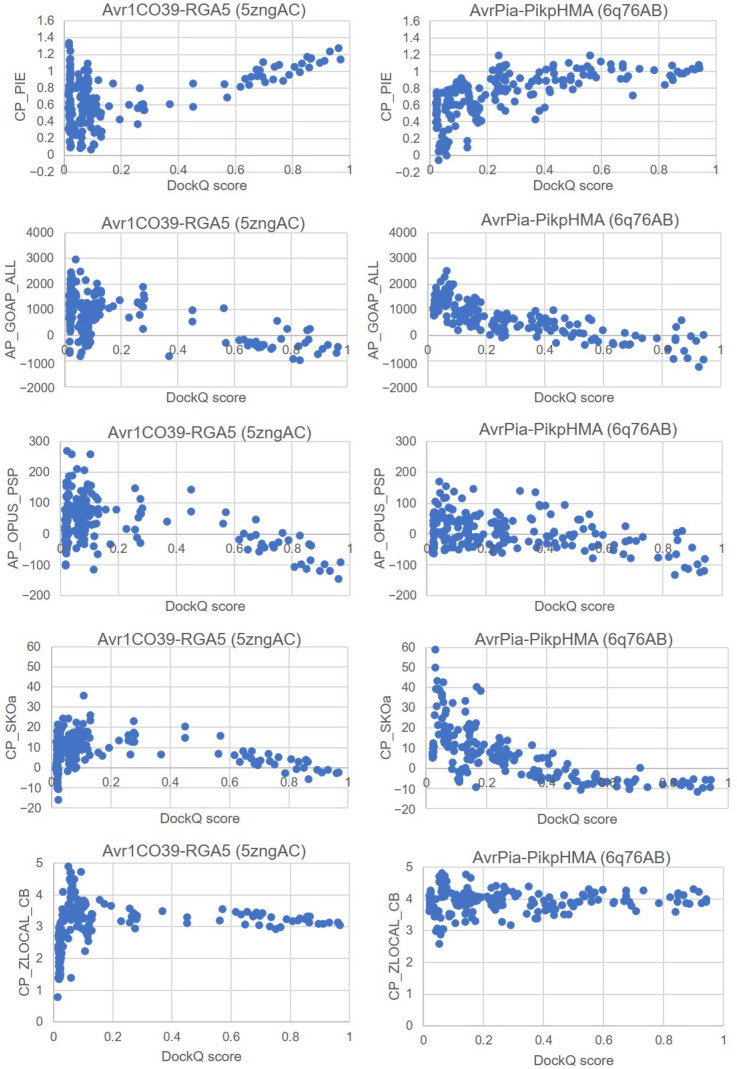
Correlation between the selected CCharPPI scoring function and DockQ scores of the top-200 docking poses using ZRANK scores for MAX region 2 complexes: Avr1CO39-RGA5 (PDB ID 5zngAC) (**left column**) and AvrPia-PikpHMA (PDB ID 6q76AB) (**right column**). The selected CCharPPI scoring functions were CP-PIE, AP_GOAP_ALL, AP_OPUS_PSP, CP_SKOa, and CP_ZLOCAL_CB.

**Figure 7 ijms-24-15239-f007:**
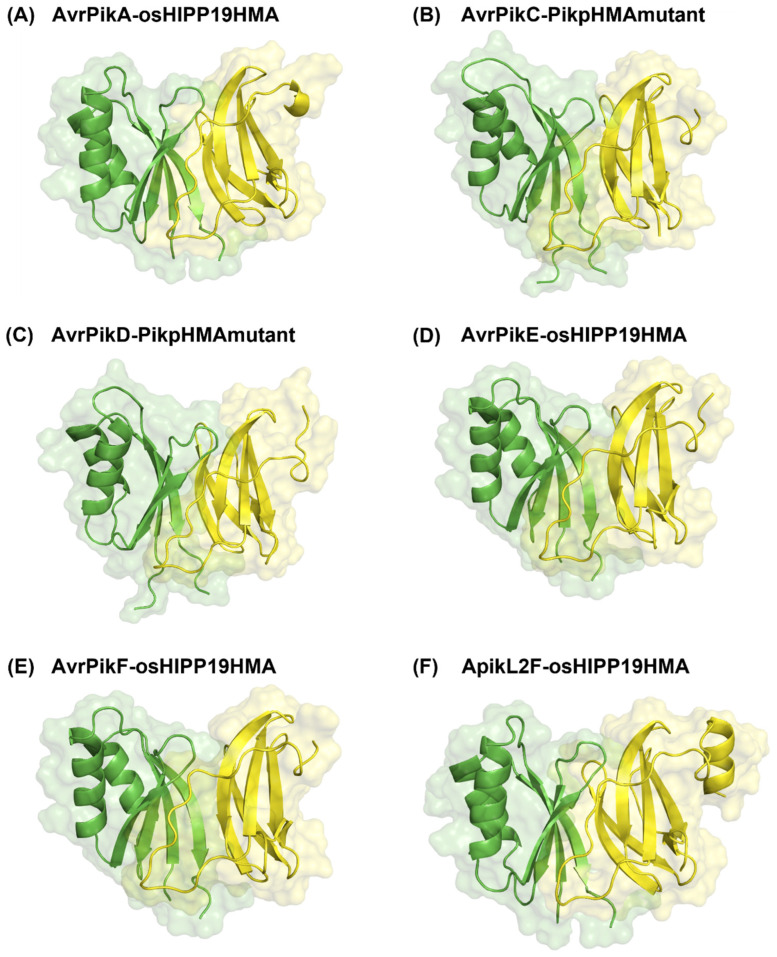
Top docking poses predicted by unbound docking complexes (**A**) AvrPikA-osHIPP19, (**B**) AvrPikC-PikpHMAmutant, (**C**) AvrPikD-PikpHMAmutant, (**D**) AvrPikE-osHIPP19, (**E**) AvrPikF-osHIPP19, and (**F**) ApikL2F-osHIPP19. Plant HMA-domain containing proteins are shown in green and effector proteins are shown in yellow.

**Figure 8 ijms-24-15239-f008:**
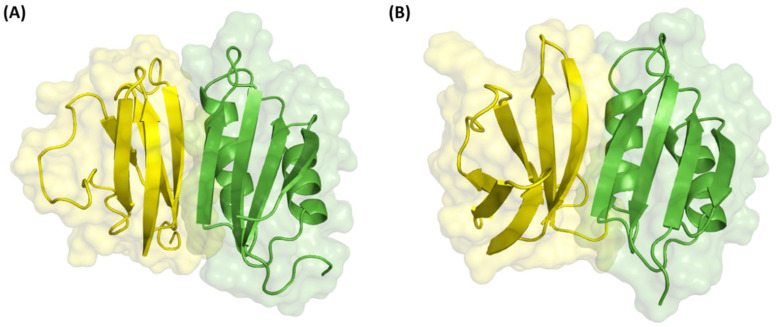
Top docking poses predicted by unbound docking of complexes (**A**) AvrPia-RGA5 and (**B**) Avr1CO39-PikpHMA. Plant HMA-domain containing proteins are shown in green and effector proteins are shown in yellow.

**Table 1 ijms-24-15239-t001:** Evaluation of ZDOCK predictions for 54,000 docking poses of bound complexes based on DockQ score. Docking poses with both the best DockQ score and the best ZDOCK score ranked as top pose are highlighted in bold. Docking poses with a DockQ score below 0.5 are in italics.

Effector–Host Receptor Binding Partner Complex	PDB ID	Pose with Best DockQ Score			Pose with Best ZDOCK Score		
		DockQ Score	ZDOCK		ZDOCK Score	DockQ	
			Score	Rank		Score	Rank
**Avr1CO39-RGA5**	**5zngAC**	**0.967**	34.19	756	46.35	*0.022*	1480
AvrPia-PikpHMA	6q76AB	0.962	42.47	3	43.46	*0.024*	1605
AvrPikC-PikhHMA	7a8xBC	0.971	60.16	2	61.74	0.910	2
	7a8xEF	**0.919**	**73.13**	**1**	**73.13**	**0.919**	**1**
AvrPikD-PikmHMA	6fu9AB	0.942	59.69	9	69.34	0.833	13
	6fu9CD	0.941	62.21	3	63.39	0.915	2
AvrPikE-PikmHMA	6fubAB	0.935	58.90	4	64.49	0.925	2
AvrPikE-PikpHMA	6g11BC	0.908	51.14	29	65.69	0.885	5
	6g11EF	0.947	58.82	4	62.28	0.753	34
	6r8mFG	0.924	63.82	2	67.90	0.901	2
AvrPikF-OsHIPP19	7b1iBC	0.928	63.75	9	75.98	0.887	8
ApikL2F-sHMA94	7nmmAI	0.952	62.99	3	68.51	0.922	2
	7nmmBJ	0.940	53.48	12	59.6	0.876	8
	7nmmCK	0.933	61.34	3	65.57	0.858	9
	7nmmDL	**0.943**	**70.17**	**1**	**70.17**	**0.943**	**1**
	7nmmEM	0.922	59.95	5	62.04	0.831	18
	7nmmFN	0.933	61.17	4	66.76	0.866	9
	7nmmGO	0.941	64.19	2	64.39	0.913	2
	7nmmHP	0.914	57.23	8	63.64	0.890	5

**Table 2 ijms-24-15239-t002:** Evaluation of ZRANK predictions for the top 2000 docking poses based on DockQ score. Docking poses with both the best DockQ score and the best ZRANK score ranked as top pose are highlighted in bold. Docking poses with a DockQ score below 0.5 are in italics.

Effector–Host Receptor Binding Partner Complex.	PDB ID	Pose with Best DockQ Score			Pose with Best ZRANK Score		
		DockQ Score	ZRANK		ZRANK Score	DockQ	
			Score	Rank		Score	Rank
**Avr1CO39-RGA5**	**5zngAC**	**0.967**	−74.5563	35	−89.5292	0.912	8
AvrPia-PikpHMA	6q76AB	0.962	−64.0926	212	−95.9139	*0.24*	204
AvrPikC-PikhHMA	7a8xBC	**0.971**	**−88.0183**	**1**	**−88.0183**	**0.971**	**1**
	7a8xEF	0.919	−85.0809	25	−109.556	0.904	4
AvrPikD-PikmHMA	6fu9AB	0.942	−92.2094	19	−120.033	0.833	13
	6fu9CD	0.941	−100.436	14	−129.325	*0.12*	296
AvrPikE-PikmHMA	6fubAB	0.935	−90.305	15	−106.179	*0.162*	239
AvrPikE-PikpHMA	6g11BC	0.908	−76.3809	31	−110.703	0.892	4
	6g11EF	0.947	−79.1564	22	−96.3179	0.875	6
	6r8mFG	0.924	−61.0644	254	−112.655	0.807	13
AvrPikF-OsHIPP19	7b1iBC	0.928	−66.7719	94	−125.125	0.88	10
ApikL2F-sHMA94	7nmmAI	**0.952**	**−103.174**	**1**	**−103.174**	**0.952**	**1**
	7nmmBJ	**0.94**	**−102.48**	**1**	**−102.48**	**0.94**	**1**
	7nmmCK	0.933	−88.5095	14	−102.381	0.834	15
	7nmmDL	0.943	−100.989	3	−104.301	0.931	2
	7nmmEM	0.922	−89.2384	14	−98.7345	*0.065*	413
	7nmmFN	**0.933**	**−107.157**	**1**	**−107.157**	**0.933**	**1**
	7nmmGO	0.941	−96.1059	4	−107.128	0.901	5
	7nmmHP	0.914	−91.2643	10	−103.77	*0.061*	522

**Table 3 ijms-24-15239-t003:** DockQ scores of docking poses with the best ZDOCK and ZRANK scores (top 2000 docking poses). The docking pose with significant change in DockQ score (i.e., DockQ score becoming >0.8) from the use of either ZDOCK or ZRANK scores is highlighted in bold. Docking poses with DockQ scores below 0.5 are highlighted in italics.

Effector–Host Receptor Binding Partner Complex	PDB ID	DockQ Score	
		Best Docking Pose ZDOCK Score 54k and 2k	Best Docking Pose ZRANK Score 2k
**Avr1CO39-RGA5**	5zngAC	*0.022*	**0.912**
AvrPia-PikpHMA	6q76AB	*0.024*	**0.240**
AvrPikC-PikhHMA	7a8xBC	0.910	0.971
	7a8xEF	0.919	0.904
AvrPikD-PikmHMA	6fu9AB	0.833	0.833
	6fu9CD	**0.915**	*0.120*
AvrPikE-PikmHMA	6fubAB	**0.925**	*0.162*
AvrPikE-PikpHMA	6g11BC	0.885	0.892
	6g11EF	0.753	0.875
	6r8mFG	0.901	0.807
AvrPikF-OsHIPP19	7b1iBC	0.887	0.880
ApikL2F-sHMA94	7nmmAI	0.922	0.952
	7nmmBJ	0.876	0.940
	7nmmCK	0.858	0.834
	7nmmDL	0.943	0.931
	7nmmEM	**0.831**	*0.065*
	7nmmFN	0.866	0.933
	7nmmGO	0.913	0.901
	7nmmHP	**0.890**	*0.061*

**Table 4 ijms-24-15239-t004:** Total number of poses retained and their DockQ scores during each step of CCharPPI filter application for complexes of MAX region 2.

CCharPPI Scoring Functions		Avr1CO39-RGA5 (5zngAC)			AvrPia-PikpHMA (6q76AB)		
		Total Poses	DockQ < 0.5	DockQ > 0.5	Total Poses	DockQ < 0.5	DockQ > 0.5
Positive filter	(1) CP_PIE > 0.8	76	48	28	81	48	33
	(2) AP_GOAP_ALL < 0	39	15	23	29	5	23
	(3) AP_OPUS_PSP < −100	7	1	6	5	0	5
Negative filter	(4) CP_SKOa > 0	6	0	6	5	0	5
	(5) CP_ZLOCAL_CB < 3	6	0	6	5	0	5

**Table 5 ijms-24-15239-t005:** Ranking of the top pose after the unbound docking of effectors binding to plant HMA proteins through MAX complex region 1 based on the use of the ZDOCK50/ZRANK-80 filter. Best ZRANK scores below −80 are shown in italics. Poses that were top-ranked by both ZDOCK and ZRANK scores are highlighted in bold.

Host Receptor Binding Partner/Effector	AvrPikA (6fubD)				AvrPikC (7a8xC)			
	Rank	ZDOCK Score	Rank	ZRANK Score	Rank	ZDOCK Score	Rank	ZRANK Score
PikmHMA/6fu9	3	60.52	2	−106.805	22	51.30	14	−88.187
PikhHMA/7a8x	*10*	*51.64*	*21*	*−79.4618*	15	54.24	1	−114.65
PikpHMA/5a6w	16	51.31	1	−91.1659	10	51.82	3	−87.068
PikpHMA-mut/6r8m	3	61.01	3	−92.2020	**1**	**63.85**	**1**	**−111.540**
ancHMA/7bnt	8	51.34	5	−92.3646	*4*	*53.23*	*89*	*−67.468*
osHIPP19/7b1i	1	71.15	2	−107.887	11	57.32	5	−94.782
sHMA94/7nmm	19	50.38	39	−81.2881	5	54.58	4	−93.776
	ApikL2F (7nmmI)				AvrPikD (5a6wC)			
PikmHMA/6fu9	3	57.53	17	−82.0646	5	62.53	2	−101.76
PikhHMA/7a8x	2	54.29	1	−92.6857	9	55.61	2	−113.11
PikpHMA/5a6w	*1*	*56.14*	*13*	*−78.5247*	13	56.28	1	−116.840
PikpHMA-mut/6r8m	11	51.88	6	−81.2988	**1**	**71.46**	**1**	**−115.93**
ancHMA/7bnt	1	55.86	3	−96.2088	1	65.07	3	−104.54
osHIPP19/7b1i	2	69.92	1	−100.196	12	60.54	1	−116.76
sHMA94/7nmm	**1**	**66.00**	**1**	**−100.020**	2	68.34	1	−112.6
	AvrPikE (6g11C)				AvrPikF (7b1iC)			
PikmHMA/6fu9	53	50.43	1	−113.286	8	57.30	1	−103.87
PikhHMA/7a8x	2	57.46	1	−104.541	*4*	*55.99*	*19*	*−76.509*
PikpHMA/5a6w	12	52.19	1	−98.2134	15	51.03	1	−92.64
PikpHMA-mut 6r8m	8	62.33	1	−112.241	6	61.70	1	−102.24
ancHMA/7bnt	6	54.53	1	−94.1270	7	52.77	4	−83.268
osHIPP19/7b1i	3	72.65	1	−128.358	4	70.92	1	−125.13
sHMA94/7nmm	4	61.76	1	−104.071	4	62.12	1	−96.3480

**Table 6 ijms-24-15239-t006:** Percentage of interacting residues on both effector and host receptor binding partner proteins based on available experimental data for MAX region 1 complexes. The figure for unbound docking corresponds to the predicted total number of residues that form inter-molecular interactions. Known figures correspond to the total number of residues that form inter-molecular interactions based on direct evidence for interacting residues in the respective effector/host receptor binding partner proteins in their original complexed form (with a different effector). Percentages are calculated based on the number of interacting residues between the unbound docking predictions and the known total number of interacting residues.

Unbound Complexes	Number of Interacting Residues on the Effector		Number of Interacting Residues on the Host Receptor Binding Partner	
	Known	Predicted	Known	Predicted
AvrPikA-osHIPP19	28	27 (87.5%)	21	21 (100%)
AvrPikC-PikpHMAmutant	21	21 (100%)	23	23 (100%)
AvrPikD-PikpHMAmutant	30	26 (86.6%)	21	21 (100%)
AvrPikE-osHIPP19	27	25 (92.5%)	24	24 (100%)
AvrPikF-osHIPP19	29	29 (100%)	21	20 (95%)
ApikL2F-osHIPP19	24	24 (100%)	21	21 (100%)

**Table 7 ijms-24-15239-t007:** Percentage of interacting residues on both effector and host receptor binding partner proteins based on available experimental data for MAX region 2 complexes. The figure for unbound docking corresponds to the predicted total number of residues that form inter-molecular interactions. Known figures correspond to the total number of residues that form inter-molecular interactions based on direct evidence for interacting residues in the respective effector/host receptor binding partner proteins in their original complexed form. Percentages are calculated based on the number of interacting residues between the unbound docking predictions and the known total number of interacting residues.

Unbound Complexes (PDB ID)	Number of Interacting Residues on the Effector		Number of Interacting Residues on the Host Receptor Binding Partner	
	Known	Predicted	Known	Predicted
AvrPia (6q76B)-RGA5 (5zngA)	10	10 (100%)	13	10 (76.9%)
Avr1CO39 (5zngC)-PikpHMA (6q76A)	8	8 (100%)	11	11 (100%)

## Data Availability

Not applicable.

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
