# Peer review of "The Molecular Docking of MAX Fungal Effectors with Plant HMA Domain-Binding Proteins"

_ijms, 2023, doi:10.3390/ijms242015239_

Round 1

Reviewer 1 Report (New Reviewer)

The manuscript  authored by Rozana et. al., used standard approaches and thoroughly addressed the limitations associated with molecular docking of protein -protein interfaces. I am inclined to accept your manuscript after addressing one minor comment. I am curious why the authors did not explore the flexible docking techniques in their study, particularly given its importance in sampling of side-chain conformations at the protein-protein interface?

Author Response

We agree with the reviewer that it is of interest to investigate the role of conformational changes through the use of flexible docking. We are indeed currently finalising a separate study using precisely this approach, which will be the subject of a separate publication.

Reviewer 2 Report (Previous Reviewer 2)

This version of the manuscript is much improved compared to the original submission. It is obvious that the authors did major efforts to adress all of the reviewers'concerns, providing convincing. This is the reason why I am supportive to publish this manuscript in BJ.

Author Response

We thank the reviewer for approving our manuscript.

This manuscript is a resubmission of an earlier submission. The following is a list of the peer review reports and author responses from that submission.

Round 1

Reviewer 1 Report

Enjoyed reading it. I don’t have further comments.  

No further comments 

Reviewer 2 Report

see attached pdf

There are some typos place to place... This can be easily corrected by proof-reading after shortening the manuscript.

Reviewer 3 Report

This is a very heavy computational assessment of effector proteins but the manuscript does not describe from which fungal species or plant species. There is no context. Is there any experimental evidence to back up the claims of this research since it is entirely computational and to my ready lacks any insights in " real biology". Effector proteins are in fact difficult to assess or predict even though, as the manuscript read, there are programs that can "predict" these complexes but in what real systems/

English is ok